# Rethinking Suicide in Rural Australia: A study Protocol for Examining and Applying Knowledge of the Social Determinants to Improve Prevention in Non-Indigenous Populations

**DOI:** 10.3390/ijerph16162944

**Published:** 2019-08-16

**Authors:** Scott J. Fitzpatrick, Bronwyn K. Brew, Donna M. Y. Read, Kerry J. Inder, Alan Hayes, David Perkins

**Affiliations:** 1Centre for Rural and Remote Mental Health, The University of Newcastle, PO Box 8043, Orange East, NSW 2800, Australia; 2Centre for Big Data Research in Health and National Perinatal Epidemiology and Statistics Unit, School of Women and Children’s Health, University of New South Wales, Level 4, Lowy Cancer Research Centre, Cnr High & Botany St, Kensington, NSW 2052, Australia; 3School of Nursing and Midwifery, University of Newcastle, Callaghan, NSW 2308, Australia; 4Family Action Centre, University of Newcastle, Callaghan, NSW 2308, Australia

**Keywords:** suicide, suicide prevention, rural Australia, social determinants, intersectionality, public health, mixed methods

## Abstract

Disproportionate rates of suicide in rural Australia in comparison to metropolitan areas pose a significant public health challenge. The dynamic interrelationship between mental and physical health, social determinants, and suicide in rural Australia is widely acknowledged. Advancement of this knowledge, however, remains hampered by a lack of adequate theory and methods to understand how these factors interact, and the translation of this knowledge into constructive strategies and solutions. This paper presents a protocol for generating a comprehensive dataset of suicide deaths and factors related to suicide in rural Australia, and for building a program of research to improve suicide prevention policy and practice to better address the social determinants of suicide in non-indigenous populations. The two-phased study will use a mixed-methods design informed by intersectionality theory. Phase One will extract, code, and analyse quantitative and qualitative data on suicide in regional and remote Australia from the National Coronial Information System (NCIS). Phase Two will analyse suicide prevention at three interrelated domains: policy, practice, and research, to examine alignment with evidence generated in Phase One. Findings from Phase One and Two will then be integrated to identify key points in suicide prevention policy and practice where action can be initiated.

## 1. Introduction

Recent data on suicide in Australia indicates that, at 15.3 deaths per 100,000 people, the rate of suicide for areas outside of capital cities is over 50% higher than in capital cities [1]. Geographic variation in patterns of suicide rates, both nationally and internationally, indicate that social factors are likely to contribute as much as, if not more than, individual factors [2]. For example, social environments that feature high rates of instability, unemployment, and material disadvantage are associated with high rates of mental distress [3]. Relative socioeconomic disadvantage has also been linked with suicide, with widening socioeconomic inequalities associated with increases in suicide rates among the most socially disadvantaged [4,5].

A major unresolved problem for suicide prevention is the difficulty of integrating social determinants frameworks into policy and practice [6,7]. In 2008, the World Health Organization (WHO) assembled and assessed the evidence for the social determinants of health and provided key recommendations for action in its report ‘Closing the Gap in a Generation: Health Equity Through Action on the Social Determinants of Health’ [8]. Yet, in a subsequent report on suicide, ‘Preventing Suicide: A Global Imperative’, there was no mention of the importance of addressing social determinants in the list of typical components for the creation of comprehensive national suicide prevention strategies [9]. 

In Australia, current suicide prevention strategies largely exclude the social determinants of suicide in favor of a multicomponent ‘systems approach’ that is directed toward improving the coordination of existing suicide prevention interventions such as the treatment of those with mental health problems, gatekeeper training, community and school-based education, and responsible media reporting of suicide [10]. These interventions are important, and current strategies provide much needed coordination and integration of existing services. However, social, political, economic, and cultural factors unique to rural communities, including regional and remote areas of Australia, that impact mental health and suicide often fall outside of these frameworks [6].

The significance of social factors corresponds with lay perceptions of risks for suicide in rural communities. Rural youth talk about unemployment, poor economies, and local norms [11]. Farmers talk about government legislation and regulation in land and water use [12]. Indigenous communities talk about forced relocation, intergenerational trauma, and a diminished sense of cultural identity [13]. 

Evidence of the importance of the social determinants of mental health and suicide in Australian rural communities has accumulated with research identifying issues related to employment and financial adversity [14], equity of access to services [15], rural masculinities [16], and climate change and adversity [17]. While this research has improved understanding of the social determinants contributing to suicide risk among rural populations, it does not account for the complex ways in which multiple determinants of suicide intersect, or for heterogeneity within social groups or across rural Australia. Moreover, there has been minimal critical examination of the implications of these findings for the planning, implementation, and evaluation of suicide prevention policy and practice. 

Related to this is the more general problem of conceptualising and measuring how mental and physical health, social determinants, and suicide in rural areas interact. Suicide prevention experts have long recognised the need to move beyond decontextualised analyses of specific risk factors for suicide to understand how different but interdependent and reinforcing factors contribute to suicide [7]. However, while age-, sex- and gender-based analyses and social determinants perspectives have been applied to research on suicide, these methods are limited by a lack of explicit theory for understanding the interrelatedness of diverse proximal (individual) and distal (social) factors. Societal and individual factors do not partition neatly across levels and, hence, suicide is irreducible to any one contributing factor—a common problem in biomedical and sociological approaches that presuppose a divide between proximal and distal factors [18]. Without an understanding of the interrelatedness of individual and social risk factors for suicide, traditional specialist mental health models and the suicide prevention strategies that support them, including mental health education and gatekeeper training programs, may be culturally unresponsive [19].

Persistent and increasing inequalities in suicide are challenging researchers to move beyond existing approaches and to engage in broader frameworks of analysis that are able to assimilate complexity. A starting place for this work is ‘intersectionality’. Intersectionality is an important theoretical framework for public health due to its focus on multiple social factors and interlocking systems of social and health inequities [20,21]. Intersectionality has been used in studies of mental health, disability, dementia, trauma, and violence, and a number of practical guides have been developed to inform public health services research, practice, and policy making [21]. 

Health-supporting public policies that address the social determinants of suicide such as education, housing, vocational training, and employment indicate an aspiration on the part of governments to achieve integrated solutions across sectors [22]. Given the National Mental Health Commission’s [23] aim of reducing suicide rates by 50% over the next decade and current moves to rebalance investment and identify medium-term change goals in national suicide prevention [24], a new approach is needed that examines the interrelationships between mental and physical health, social determinants, and suicide in rural areas. Focusing attention on under-theorised and under-researched relationships between factors using an intersectionality informed mixed-method approach will address an important gap in evidence about the social determinants of suicide, and identify new intervention points to improve suicide prevention in rural Australia.

## 2. Research Design, Methodology, and Methods

### 2.1. Research Questions

This research addresses this gap about the social determinants of suicide by answering the overarching research questions:How do the social determinants of health interact with physical and mental health and suicide in rural Australia?What social processes or circumstances contribute to the interaction between social determinants, physical and mental health, and suicide in rural Australia?To what extent do suicide prevention policies align with the evidence about suicide in rural Australia?What can be done to improve suicide prevention policy and practice to better address the social determinants of suicide in rural Australia?

In answering these questions, coronial data on intentional self-harm deaths from regional and remote Australia from the National Coronial Information System (NCIS) will be collected and analysed. These data will be compared with four national and four state policy documents to examine processes of priority setting and resource allocation in suicide prevention, and the extent to which the policies align with the needs of individuals and groups who contribute a disproportionate share to the burden of suicide. In doing so, this research will generate a more substantive understanding of suicide that combines demographic, psychological, and psychiatric variables with social, cultural, and political factors to improve suicide prevention strategies for rural populations. 

The study will be carried out in two consecutive phases to align with research questions. A mixed methods study design informed by the theoretical underpinnings of intersectionality will be used [25] (see Figure 1).

### 2.2. Theoretical Framework

Intersectionality is both a theoretical and methodological approach for improving understanding of, and responsiveness to, diversity in health and illness [21]. Emerging from the works of critical social science scholars, intersectionality seeks to elucidate the multifaceted nature of experience by viewing social categories such as race, class, and gender as deeply intertwined. Broadened by public health researchers to include a range of other aspects of social position such as age, sexuality, indigeneity, geography, (dis)ability, and socio-economic status, intersectional approaches focus on the interrelationship between different factors and the hybrid positions and identities associated with them [26,27].

As well as providing a pragmatic tool for theorising the complex relationship between different factors, intersectionality can help yield greater insights into the combined impacts of social and structural processes and circumstances that produce particular relationships between factors that influence health [26,28]. Links between rural health disparities and government policies in areas of Indigenous Affairs, health, welfare, and primary industries have been documented, yet existing approaches have been unable to capture how these inequalities are shaped by social, political, and economic processes [29]. Intersectionality responds to this problem by providing a more nuanced understanding of health determinants that underscores the impacts of policies and practices on different groups [29].

### 2.3. Phase One—Study of Coronial Suicide Data from Regional and Remote Australia

#### 2.3.1. Data Collection

From November 2016 to June 2019, all closed cases of intentional self-harm deaths in regional and remote Australia across four jurisdictions (New South Wales, Queensland, South Australia, and Tasmania) for the period January 2010–October 2016 were extracted from the National Coronial Information System (NCIS) (*n* = 3761). These jurisdictions were chosen due to availability of data and time needed to obtain additional agreements and approvals from local research ethics committees to access data from Victoria and Western Australia. This dataset was compiled by the Centre for Rural and Remote Mental Health, The University of Newcastle, Australia. Cases were identified using ICD-10 (International Statistical Classification of Diseases and Related Health Problems-Tenth Revision) cause of death codes for intentional self-harm as classified by the WHO [30], and calculated by postcode using Australian Statistical Geography Standard (ASGS) Remoteness Area Codes [31]. Both population- and individual-level data were collected. This includes coded data from the NCIS on: (a) case demographics (year of birth, postcode of usual residence, ASGS-remoteness factor, sex, marital status, indigeneity, employment status, usual occupation, country of birth, period of residency in Australia); and (b) cause of death details (time, location, and postcode of incident and death, mechanism and object or substance producing injury) (see Appendix A).

Social determinants data were coded from case reports (police, autopsy, toxicology reports, and coroner’s findings) for each individual case to enable both quantitative and qualitative analysis. Quantitative coding and classification drew upon ICD-10 Z-Codes for health hazards relating to socioeconomic and psychosocial circumstances [31], with some minor modifications made to streamline data management and capture information on risks that are well-known in the literature on suicide such as death of a family member or friend by suicide [32]. This included information about housing and economic circumstances, family violence, legal issues, interpersonal relationships and supports, and so forth (see Appendix A). Additional codes were developed with specific relevance to suicide and other important factors such as alcohol and/or other drug use, service contacts, and any physical and mental health history (see Appendix A). All available case reports for suicides for the period 2010–2016 have been downloaded from the NCIS. The case report data will be the basis for the qualitative component of Phase One.

#### 2.3.2. Data Analysis 

The analyses will help to identify the linkages between social determinants, mental and physical health, and other situational data available from the dataset, and to capture the complexity of the relationships between them.

Quantitative analysis: From the above dataset, two methods will be used to identify intersections or clusters in the suicide population. The first will use cross-tabulation and statistical interaction to identify intersections of up to four risk variables. The second will use a machine learning approach to identify the main clusters with any number of risk variables according to the best model fit [33].

The first method will generate univariate and bivariate proportions in the suicide population for each of the demographic and social determinant variables. Chi-squared analyses will be used to identify pairs of variables that are statistically likely. We will then test identified triple and quadruple combinations statistically by adding interaction terms to logistic regression models [34]. Since there is no control group, we will randomly allocate one variable from the original pair to be an exposure variable and the other to be the outcome variable for modelling purposes. If the interactions are significant, further stratification of the third and fourth variables will give the strength of association between three or four variables; that is, the statistical likelihood that these variables occur together in regional and remote suicide. Results will be reported both as additive and multiplicative interaction. As it is already known that significant differences exist between male and female suicide and suicide within older and younger cohorts [35], analyses will be stratified by sex and age. 

The second method will employ latent class analysis, a machine learning approach that uses non-parametric data that is hypothesis free, to identify clusters or groups within the data. This method has been successfully applied to suicide data as it does not require a control group [36,37]. All demographic and social determinant variables will be entered into the latent class model (using MPLUS) and Bayesian information criteria will be used to determine the number of clusters best fitting the data. Results are expressed as conditional probabilities which will then be used by the investigators to determine labels and descriptions for each cluster. For each intersection group and cluster identified from methods one and two, the average level of treatment and access to services will be determined in order to highlight groups that may be more vulnerable and less easy to access. 

Qualitative Analysis: Drawing on case report data (police, autopsy, toxicology reports, and coroner’s findings) and informed by the quantitative data analysis, a sample of approximately 50 cases will be selected for in-depth qualitative analysis. A qualitative intersectional analysis will provide a theoretical framework for examining statistically significant interactions in greater depth. Data contained in case reports varies, and the amount and quality of data will inevitably determine decisions about which cases to analyse. Where quantitative analysis reveals the intersection of multiple factors, data-rich case reports will be selected for subsequent qualitative analysis. 

Using McCall’s [38] framework for the operationalisation of intersectionality, the study will employ an intercategorical approach. The intercategorical approach involves the provisional use of analytic categories, although these categories are not treated as static or as additive, but as dynamic and multiplicative. The intercategorical approach posits that multiple analytic categories lead to the formation of more detailed social groups to empirically examine suicide mortality in rural Australia in greater depth. A deductive approach using an analytic template will be applied to the data. This approach involves: (i) analyses of interacting social categories and determinants of suicide within an individual case or other dimensions of the case that interact with relevant social categories/determinants; and (ii) the drawing of connections between social categories/determinants and broader social and structural relations that shape experience [39,40].

### 2.4. Phase Two—Policy Analysis

Intersectional methods will guide the analysis of eight key documents. The first of these, ‘The Fifth National Mental Health and Suicide Prevention Plan’ [41], seeks to establish a national approach for collaborative government efforts in the area of mental health and suicide prevention; the second is Suicide Prevention Australia’s ‘Transforming Suicide Prevention Research: A National Action Plan’ [24] that seeks to deliver a blueprint for Australian suicide prevention research agenda setting and funding; the third is the National Mental Health Commission’s ‘National Report on Mental Health and Suicide Prevention’ [42] that reports on outcomes of engagement with key stakeholders; and the fourth is the Senate Community Affairs Reference Committee’s report ‘Accessibility and Quality of Mental Health Services in Rural and Remote Australia’ [43] that develops a list of recommendations from the committee’s 2018 inquiry. In addition, current state suicide prevention strategies for Queensland [44], New South Wales [22], South Australia [45] and Tasmania [46] will be analysed. These documents deserve careful analysis because they establish both the foundation for key components of local and national suicide prevention strategies and a vision for transformational change. 

The applied nature of suicide prevention means that both empirical and normative aspects are integrated within its field of practice. That is to say, suicide prevention is concerned with acquiring knowledge about suicide, as well as applying that knowledge to reduce harms [47]. The policy analysis will, therefore, examine suicide prevention at three interrelated domains: policy, practice, and research. First, the extent to which current suicide prevention strategies address the needs of those rural populations identified in Phase One of the study will be examined. Second, the normative elements that guide suicide prevention policy and practice including the role of evidence and processes by which decisions are made and evaluated will be analysed. Third, factors that may constrain or support effective interventions, including social, political, economic, legal, and cultural contexts will be examined [48]. Finally, points in suicide prevention policy and practice where change can be initiated to address the social determinants of suicide will be identified.

### 2.5. Ethical Considerations

The study has been approved by the Justice Human Research Ethics Committee (Reference Number CF/16/14124) and The University of Newcastle Human Research Ethics Committee (Protocol Number H-2016-0252). All data will be treated confidentially and the privacy and integrity of individuals respected both during the research process and in the dissemination of results.

Data on suicide among Aboriginal and Torres Strait Islander peoples is available through the NCIS and will be included in overall statistical analysis. However, personal information from case reports in cases involving Aboriginal and Torres Strait Islander peoples have not been extracted and will not be analysed. Significant differences exist between suicide in Aboriginal and non-Aboriginal populations with regards to risk factors and responses to suicide prevention programs. Accordingly, some researchers argue for Aboriginal suicidology to be separate from current mainstream suicidology [49,50]. The limited scope and duration of this study means that it is not equipped to provide a comprehensive, sensitive, and nuanced study of suicide among Aboriginal and/or Torres Strait Islander peoples with respect to its historical, cultural, socio-political, and situational complexity. 

## 3. Discussion

### 3.1. How do Mental and Physical Health, Social Determinants, and Suicide in Rural Australia Interrelate?

To date, analyses of inequalities in suicide mortality have largely focused on a single social category such as gender, rurality, indigeneity, mental health, sexuality, unemployment, occupational status or socio-economic status. Similarly, suicide prevention activities often ‘target’ at-risk groups based on single factors or identities (for example: men, rural and remote, youth, indigenous peoples, and lesbian, gay, bisexual, transgender, intersex, and queer communities (LGBTIQ+)). Narrow conceptualisations, however, are problematic as there is considerable diversity within groups and communities and between places [29]. Certain categories of difference may also be naturalised. That is, where culturally specific worldviews (such as those around gender or race) are perceived within a culture as normal, self-evident and, consequently, as universal and absolute [51]. Naturalised categories invariably lend themselves to homogenisation and a ‘one-size-fits-all’ approach that does not adequately reflect lived reality [26].

The category of ‘rural’ is subject to naturalisation through social and cultural representations that depict, in various ways, the landscape, inhabitants, culture, and ways of life. Often produced in dichotomous relationship to ‘urban’, such representations of rurality influence perceptions of rural communities and rural life that are important in constructing suicide in rural areas as a problem and developing appropriate prevention strategies. Population level data that shows suicide rates increasing with remoteness, together with generalisations about people living in rural areas as stoic, less health literate, and disengaged from health services, shape perceptions of rural communities and rural health consumers as homogenous, uneducated, and not actively involved in health care [52]. ‘Rural’ however, is a heterogeneous concept [53]. Rural health consumers are diverse, while areas categorised as rural experience clear differences in social, economic, and health outcomes. Spatial analyses of suicide in Australia, for example, show considerable variation across rural communities, highlighting the importance of contextual or place-based factors [53,54,55]. 

The large body of research that shows how suicide is socially distributed, and that underscores the sociocultural, political, and economic conditions that foster poor mental health and suicide, indicates that people can be distressed and suicidal without being psychiatrically disordered [56,57]. Even when serious and persistent mental disorders are acknowledged as increasing a person’s vulnerability to suicide, it does not necessarily follow that separating mental illness from an individual’s own experience of it helps the understanding of suicide [58]. It may be the case that a range of associated problems such as housing, unemployment, or social exclusion are more important precipitating factors in the event of a suicide than those prescribed by a purely biomedical reading. Moreover, the classification of symptoms without regard to aetiology in current medical practice means that many of the social factors that precipitate symptoms of depression, anxiety, and grief are not distinguished from those produced by some form of individual pathology. In such cases, it is not clear whether medical treatment produces better or more effective responses than social support or political action that directly targets the sources of this distress [19,59].

The study methodology in this protocol addresses reductionist assumptions about suicide in several ways. First, by the inclusion of categories such as employment status, occupation, mental health diagnosis, physical disability and geographical area, which are then analysed fully in their intersections with gender and age. By aiming to uncover how different factors converge, affect one another, and create new social locations and experiences of identity, the study moves beyond the simple assumption that suicide “*may be* caused by a number of contributing causes by asserting that numerous factors are *always* at play” [29] (p. 276). 

Second, it will extend beyond the analyses of social categories to include analyses of the social and structural processes and circumstances that produce particular relationships between determinants that impact suicide in rural Australia. Glaring inequalities in health and social outcomes are often the result of social inequities or public policies that create the conditions for increased incidence of health and social outcomes that adversely affect particular groups and communities [60]. From an intersectional perspective, a key question is: ‘How do societal norms and economic and political conditions strengthen relationships between determinants of suicide in rural Australia?’ This line of questioning leads to an explicit focus on social, political, and economic divisions that affect vulnerabilities to suicide and access to appropriate services and support [29,51].

### 3.2. How can Policy and Services Better Address These Complex Experiences?

In recognition of local differences, there has been an observable shift in suicide prevention policy in Australia to engage communities in developing local, culturally appropriate strategies that address local need [10]. Funding and service models, adequate service provision, retention and training of a mental health workforce, together with other barriers to access, such as ensuring culturally competent services and positive attitudes to mental health have been identified as key to achieving reform and improved outcomes [43]. While important for improving service development and building community capacity for suicide prevention, the focus on local action in suicide prevention strategies has led to an over-riding emphasis on ‘downstream’ approaches targeting individuals and clinical services.

Coding of individual case reports from the NCIS data during data collection provides supporting evidence for the interrelationship between mental and physical health, social determinants, and suicide in rural Australia, as hypothesised by the research team. This includes observed interrelationships between: (i) mental illness, unemployment, and drug and alcohol use; (ii) old age, physical illness, and disability; (iii) male gender, drug and alcohol use, and the perpetration of family violence. These emergent findings point to problems specific to delivering programs and services to rural communities in areas relating to (un)employment, family violence, drug and alcohol use, and aged and end-of-life care including workforce, community education, the criminal justice system, rural employment, and ineffective practice models. These, however, are not adequately developed into a clear problem by suicide prevention experts, other than a focus on high-needs groups such as men, youth, the elderly, and Aboriginal and Torres Strait Islander peoples [61]. Moreover, policies and processes that create and sustain disparities in physical and mental health, unemployment, violence, and drug and alcohol use, and their influence on suicide, are not systematically linked [29].

Barriers to addressing the social determinants include scientific uncertainty about how political decision-making can affect suicide, a focus on clinical and behavioural risk factors in health research, competing demands of different interest groups and political ideologies on the formulation of policy options, and political preferences for simple and manageable policy solutions geared for quick wins [61,62,63,64,65]. Evidence, therefore, is only one source of information in the policy making process [66]. In order for evidence to be successfully adopted and implemented, it must be linked to pragmatic policy solutions, strategies, and outcomes that can be delivered within “the siloed departmentalism of Australian government” [67] (p. 4). Researchers have suggested that because moral and ethical arguments underpin public policy, decoupling evidence from normative positions is a mistake [67,68]. Health equity is “an ethical concept with normative implications… since the term implies unfairness in the current state of distributive policies” [63] (p.153). The application of ethical concepts and frameworks may be of value in helping to construct inequalities in suicide in rural Australia as a ‘problem’ through which community demand for change can be built.

In addition to social determinants advocacy, findings from the study may inform the development of policy solutions that promote and improve the health needs of rural populations, as well as the integration of new decision-making or administrative processes [69] that change the current ‘logic’ of policy making in suicide prevention. The latter includes the use of ethical concepts and arguments for addressing widening inequalities in suicide mortality in rural Australia such as equity, participation, transparency, fair priority setting deliberation, and resource allocation.

## 4. Strengths and Limitations of the Study

The NCIS is an online repository of coronial data from Australia and New Zealand. To our knowledge, this study comprises the most comprehensive dataset of Australian or even international rural suicides ever collected. The coding and analysis of individual case reports using ICD-10 Z-Codes for health hazards relating to socioeconomic and psychosocial circumstances is a key innovation and will enable interdisciplinary mixed-methods research on suicide in rural Australia. The integration of quantitative and qualitative methods in suicide research has been identified as a key priority [70,71], yet mixed- or multiple-methods approaches to suicide continue to be under-utilised. The study uses a mixed-methods study design to produce generalisable results for larger population groups, as well as complementary context-dependent insights into the complexity of individual suicide. Together with policy analysis, the study addresses an important gap in evidence about the social determinants of suicide that has the potential to drive forward significant innovation in policy and program responses to improve suicide prevention in rural Australia.

The accuracy and quality of coronial data have been the subject of ongoing debate with researchers claiming significant underestimations in suicide deaths [72]. The range and quality of coronial data due to missing data from many case reports also varies across jurisdictions. If data is missing at random, multiple-imputation or missing-indicator methods will be applied [73,74]. For data not missing at random, discernible patterns within jurisdictions will be reported and inferences made where possible. The sampling of case studies for qualitative analysis based on the quality and amount of data is likely to create potential bias in the data. Those individuals who are socially isolated or marginalised are unlikely to be represented in case reports. This may result in the potential exclusion of some high-needs individuals or groups from analysis and subsequent policy recommendations. Professional judgements about suicide made by coroners and police are also likely to influence the sources of data documented in individual cases, with the likely prioritising of mental illness, problematic drug and alcohol use, and recent life events such as relationship issues [75]. A further limitation of the study is that it only focuses on four Australian states. It is likely that demographic differences and trends vary between states and, therefore, findings may not be generalisable. Despite these limitations, previous research indicates that coronial data retains some validity in making judgments about the social circumstances of suicide [76].

## 5. Significance

The large imbalance between rates of suicide in rural versus metropolitan settings make this research highly significant. Recent widening of socioeconomic inequalities in suicide mortality in Australia have been primarily associated with declines in suicide rates in areas of high socioeconomic status and increases in suicide rates in areas of low socioeconomic status [5]. The knowledge generated by this study has the potential to improve and refine suicide prevention strategies to make them more effective and equitable, especially among vulnerable groups and communities in rural Australia who experience a disproportionate share of the burden of suicide.

The study will contribute to methodological advancement in suicide research through the application of an intersectional framework for investigating the relationship between multiple intersecting determinants of suicide, the broader social and political contexts in which they are embedded, and the reasoning that lies behind current suicide prevention strategies. Suicide researchers have underscored the importance of theory integration within the process of research design and analysis [71,77]. As the first study operationalising intersectionality to examine suicide in Australia, this study takes a sophisticated multidisciplinary approach that incorporates empirical evidence with theoretical insights to advance understanding of the dynamic interrelationships among multiple determinants of suicide. The investigation of the ‘intersections’ between social determinants (education, involvement with law/justice, (un)employment, financial, and interpersonal factors), with physical and mental health, alcohol and other drug use, and health and social services use will contribute further to the growing body of empirical social research on suicide. 

Within the Australian context, a deeper understanding of suicide and suicide prevention in rural Australia has contemporary significance given the increasing pressure on federal, state, and local governments to show political leadership for suicide prevention in light of continued high rates. This study offers a timely opportunity to ask how we should best act to address the complex problem of suicide with the perspective of social determinants frameworks and, thus, has far reaching significance for rural Australia, as well as for national and international suicide prevention practice and policy. We believe the study will be internationally relevant and could serve as a model for studies in other countries.

## 6. Conclusions

The study addresses an important gap in evidence about the social determinants of suicide and their integration into policy and practice that has the potential to improve national suicide prevention strategies and local practices. As well as improved scientific understanding of suicide and a theoretical model that helps address its multifactorial nature, the study’s focus on previously unexamined issues in suicide prevention policy such as equity, participation, transparency, fair priority setting deliberation, and resource allocation will lead to further development of ethical concepts and their integration into suicide prevention policy.

## Figures and Tables

**Figure 1 ijerph-16-02944-f001:**
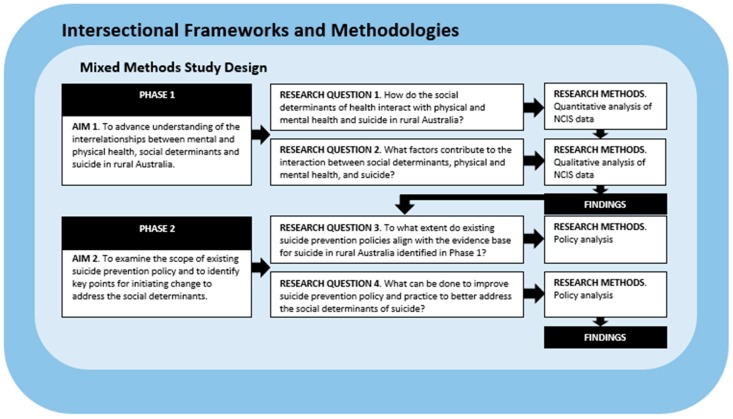
Flow of research aims and questions to methods.

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
