# Peer review of "Rethinking Suicide in Rural Australia: A study Protocol for Examining and Applying Knowledge of the Social Determinants to Improve Prevention in Non-Indigenous Populations"

_ijerph, 2019, doi:10.3390/ijerph16162944_

Round 1

Reviewer 1 Report

Comments on IJERPH 562662

Thank you for the opportunity to review this well written protocol paper outlining a follow up study that offers a new approach to the analyses of coronial data on suicide deaths. In light of the limitations of coronial data, there are a few key points that could be addressed in more detail within the text, particularly in relation to the emphasis on the perspective of social determinants frameworks.

Line 201: Data contained in case reports varies, and the amount and quality of data will inevitably determine decisions about which cases to analyse.

This approach, while sensible on a pragmatic level, creates bias in the data that I feel could be better acknowledged in the protocol.  Those cases presenting less information in Coronial records are likely to represent the most “hard to reach” socially isolated and excluded individuals, for whom the policies may not easily reach. This weakness/bias in the methodology is one of a number of well-documented limitations of using coroner’s records to understand suicide, which have received only a cursory mention in the manuscript (line 371).  The concerns associated with exclusive use of coroner’s data to understand suicide could be more robustly acknowledged.  Perhaps if these concerns are less prevalent in the Australian context, that could be clarified for international readers.

Combining the policy analyses with the findings from qualitative enquiry based on the existing data, does not in itself add to the existing data and could lead to narrow interpretations, potentially excluding high need types/groups who will be less well represented by the selection of ‘data-rich’ cases for qualitative analyses.

The additional references below may be helpful to the authors:

Discussion of limitations in understanding social and economic factors involved in suicide, when collecting data via coroner’s records:

Mallon, S., Galway, K., Hughes, L., Rondón‐Sulbarán, J., & Leavey, G. (2016). An exploration of integrated data on the social dynamics of suicide among women. Sociology of health & illness38(4), 662-675. https://doi.org/10.1111/1467-9566.12399

Discussion of limitations and changes in routine collection of data pertaining to suicide, specific to Australian context :

De Leo, D. (2015). Australia revises its mortality data on suicide.

https://doi.org/10.1027/0027-5910/a000043

Line 246: cases involving Aboriginal and Torres Strait Islander peoples have not been extracted and 246 will not be analysed.

Rural and indigenous issues and risk factors in suicide are commonly featured and studied together in the Australian context (perhaps this is not wise as the authors suggest the patterns in data are materially different), so I wondered should this exclusion be reflected in the title of the paper? Perhaps add clarity using a sub-title: in non-indigenous populations?

Line 388: The investigation of the ‘intersections’ between social determinants (education, involvement with law/justice, (un)employment, financial, and interpersonal factors), with physical and mental health, alcohol and other drug use, and health and social services use make this study unique.

It is fair to acknowledge that new statistical methods are being applied here to potentially improve the robustness of findings.  However I would not agree that the above factors have not been examined and the source of data in this protocol remains limited to Coronial data collected after the suicide event, therefore with the bias of a retrospective lens. Other studies have attempted to address this limitation by linking coronial data with lifetime primary care data and qualitative interviews with next of kin.  

e.g. Mallon S, Galway K, Rondon-Sulbaran J, Hughes L, Leavey G. When health services are powerless to prevent suicide: results from a linkage study of suicide among men with no service contact in the year prior to death. Primary Health Care Research & Development. 2019;20.

These minor points may be useful to the authors and those judged to be of relevance can be addressed relatively easily in advance of publication.  

I would like to wish the authors good luck with the study and look forward to reading the findings in due course.

Author Response

Thank you for your interest in our manuscript and for the opportunity to revise and resubmit it for publication in IJERPH. We are particularly indebted to the reviewers who read and engaged with the manuscript and provided detailed suggestions as to how it could be improved. We have addressed the specific criticisms of the reviewers individually as follows:

Reviewer 1

1. In light of the limitations of coronial data, there are a few key points that could be addressed in more detail within the text, particularly in relation to the emphasis on the perspective of social determinants frameworks.

Line 201: Data contained in case reports varies, and the amount and quality of data will inevitably determine decisions about which cases to analyse.

This approach, while sensible on a pragmatic level, creates bias in the data that I feel could be better acknowledged in the protocol.  Those cases presenting less information in Coronial records are likely to represent the most “hard to reach” socially isolated and excluded individuals, for whom the policies may not easily reach. This weakness/bias in the methodology is one of a number of well-documented limitations of using coroner’s records to understand suicide, which have received only a cursory mention in the manuscript (line 371).  The concerns associated with exclusive use of coroner’s data to understand suicide could be more robustly acknowledged.  Perhaps if these concerns are less prevalent in the Australian context, that could be clarified for international readers.

Combining the policy analyses with the findings from qualitative enquiry based on the existing data, does not in itself add to the existing data and could lead to narrow interpretations, potentially excluding high need types/groups who will be less well represented by the selection of ‘data-rich’ cases for qualitative analyses.

We agree with the reviewer. We have amended the manuscript to describe the limitations associated with the use of coronial data more explicitly, particularly the use of coronial case report data. See amended manuscript p.9 line 455-461

2. Line 246: cases involving Aboriginal and Torres Strait Islander peoples have not been extracted and 246 will not be analysed.

Rural and indigenous issues and risk factors in suicide are commonly featured and studied together in the Australian context (perhaps this is not wise as the authors suggest the patterns in data are materially different), so I wondered should this exclusion be reflected in the title of the paper? Perhaps add clarity using a sub-title: in non-indigenous populations?

We have amended the title of the manuscript as per the reviewer’s recommendation to “Rethinking suicide in rural Australia: A study protocol for examining and applying knowledge of the social determinants to improve prevention in non-Indigenous populations”. This has also been acknowledged in the abstract.

3. It is fair to acknowledge that new statistical methods are being applied here to potentially improve the robustness of findings.  However I would not agree that the above factors have not been examined and the source of data in this protocol remains limited to Coronial data collected after the suicide event, therefore with the bias of a retrospective lens. Other studies have attempted to address this limitation by linking coronial data with lifetime primary care data and qualitative interviews with next of kin.  

e.g. Mallon S, Galway K, Rondon-Sulbaran J, Hughes L, Leavey G. When health services are powerless to prevent suicide: results from a linkage study of suicide among men with no service contact in the year prior to death. Primary Health Care Research & Development. 2019;20.

We agree with the reviewer. As discussed above, we have noted the limitations associated with using coronial data. We have also revised claims made about the uniqueness of the study to acknowledge the growing body of empirical social research on suicide and how this work contributes to it. See amended manuscript p.10 line 495.

Reviewer 2 Report

I congratulate the authors on an important paper outlining a novel approach for understanding and addressing the highly complex issue of rural suicide. The proposed methods appear scientifically sound. However, I do think the authors are overly optimistic regarding the quality of the qualitative component. There is wide variation in the quality of case report data and these sources are unlikely to provide in-depth insights into the circumstances of individuals. The authors point out that it will be necessary for the amount and quality of the data to determine the inclusion of cases, which is not an ideal sampling approach. These issues should be noted in the limitations section. I recommend including an arrow in the diagram on page 4 to indicate links between the quantitative and qualitative components (as per top of page 6). The paper is well written and overall of a high standard; however, I did notice a small error - page 7, line 259, the sentence beginning 'For example' is not grammatically correct.   

Author Response

Thank you for your interest in our manuscript and for the opportunity to revise and resubmit it for publication in IJERPH. We are particularly indebted to the reviewers who read and engaged with the manuscript and provided detailed suggestions as to how it could be improved. We have addressed the specific criticisms of the reviewers individually as follows:

Reviewer 2

1. I congratulate the authors on an important paper outlining a novel approach for understanding and addressing the highly complex issue of rural suicide. The proposed methods appear scientifically sound. However, I do think the authors are overly optimistic regarding the quality of the qualitative component. There is wide variation in the quality of case report data and these sources are unlikely to provide in-depth insights into the circumstances of individuals. The authors point out that it will be necessary for the amount and quality of the data to determine the inclusion of cases, which is not an ideal sampling approach. These issues should be noted in the limitations section.

We agree with the reviewer (see also Reviewer 1 point 1). We have amended the manuscript to describe the limitations associated with the use of coronial data more explicitly, particularly the use of case report data. See amended manuscript p.9 line 455-461

2. I recommend including an arrow in the diagram on page 4 to indicate links between the quantitative and qualitative components (as per top of page 6). 

We have amended the diagram to include an arrow to indicate the link between the quantitative and qualitative components.

The paper is well written and overall of a high standard; however, I did notice a small error - page 7, line 259, the sentence beginning 'For example' is not grammatically correct.   

We have amended the paper to correct this grammatical error. See amended manuscript p.7 line 314-315.

Reviewer 3 Report

My only hesitation is that the paper is somewhat long with major points reiterated more than necessary.  But since space is not a limitation, this is a minor concern.

Author Response

Thank you for your interest in our manuscript and for the opportunity to revise and resubmit it for publication in IJERPH. We are particularly indebted to the reviewers who read and engaged with the manuscript and provided detailed suggestions as to how it could be improved. We have addressed the specific criticisms of the reviewers individually as follows:

1. My only hesitation is that the paper is somewhat long with major points reiterated more than necessary.  But since space is not a limitation, this is a minor concern.

We have revised the manuscript to remove one instance of duplication from the manuscript (see p2 line 80). Without any further editorial guidance we are reluctant to revise the manuscript further.